# When Are Neural Pruning Approximation Bounds Useful?

## Abstract

Approximation bounds for neural network pruning attempt to predict the trade-off between sparsity and fidelity while shrinking neural networks. In the first half of this paper, we empirically evaluate the predictive power of two recently proposed methods based on coreset algorithms. We identify several circumstances in which the bounds are loose or impractical to use and provide a brief analysis of the components of the bounds that contribute to those short-comings. In the second half, we examine the role of fine-tuning in prunability and observe that even tight approximation bounds would be poor predictors of accuracy after fine-tuning. This is because fine-tuning can recover large amounts of accuracy while simultaneously maintaining or increasing approximation error. We discuss the implications of these finding on the application of coreset-based pruning methods in practice and the role of approximation in the pruning community. Our code is available in the attached supplementary material.

## 1 Introduction

Recent work has attempted to prune neural networks while providing a mathematical upper bound on the deviation from the original network. These bounds were first introduced as a component in proofs of generalization in statistical learning theory (Arora et al., 2018). Recent improvements on these pruning methods have come from the field of randomized algorithms, leveraging the concept of *coresets* (Mussay et al., 2020) (Baykal et al., 2019a) (Baykal et al., 2019b) (Liebenwein et al., 2019), with empirical evaluations focusing on practical improvement in the sparsity/accuracy trade-off.

However, surprisingly little attention has been given to evaluating the tightness of the proposed bounds or the predictive power of the guarantees they provide. In the first half of this paper, we fill this gap by empirically evaluating the tightness of two recently proposed coreset bounds, using them to predict the accuracy of neural networks after pruning. We conduct a series of experiments of increasing difficulty, starting with networks that are known to be highly sparse and ending with networks in which no regularization is applied during training. We identify several circumstances in which the bounds are loose or impractical to use and provide a brief analysis of the bound components that contribute to those short-comings.

In the second half of the paper we introduce a fine-tuning procedure after pruning, which is common practice in the pruning community. Fine-tuning can drastically improve the accuracy achieved at high levels of sparsity, taking some models from 40% accuracy after pruning to 98% accuracy after fine-tuning. However, we observe that this improvement does not usually include a reduction in approximation error; indeed, fine-tuning can even *increase* approximation error in hidden layers while simultaneously recovering the original performance of the network. We argue that this phenomenon limits the predictive power of approximation bounds which are constructed "layer-by-layer" and which ignore the final softmax layer at the output. This motivates further discussion of the practical applications of coreset-based methods and whether "approximation" is the correct goal for the pruning community.

## 2 RELATED WORK

**Over-parameterization** A popular recipe for training neural networks is to make the network as large as possible such as to achieve 0 training loss, then regularize to encourage generalization (Karpathy, 2019). It is possible that such *over-parameterized* networks have have favorable optimization properties (Li et al., 2020) (Du et al., 2018b). However, over-parameterized networks are becoming increasingly burdensome to evaluate on consumer hardware, which has driven the interest in researching precise and effective pruning methods that can take advantage of sparsity in the discovered solutions.

**Pruning With Guarantees** Despite the massive literature on neural pruning techniques, few papers provide any kind of mathematical guarantee on the result of pruning. Besides the coreset methods which are discussed below, Serra et al. (2020) provide a method for "losslessly" compressing feed-forward neural networks by solving a mixed-integer linear program. They observe that applying L1 regularization during training is required to achieve non-negligible lossless compression rates, which closely aligns with the observations in our work. Havasi et al. (2018) provide an explicit compression/accuracy trade-off for Bayesian neural networks, but their compression method cannot be used to improve inference efficiency, only the memory needed to store the model.

**Empirical Scaling Laws** Frankel et al. have observed that error after pruning can be empirically modeled as a function of network size, dataset size, and sparsity with good predictive results (Rosenfeld et al., 2020). This can be seen as a more practical alternative to formal guarantees on accurancy after pruning, where a few small-scale experiments are run to estimate the accuracy after pruning at larger scales.

## 3 CORESET APPROXIMATION BOUNDS

### 3.1 CORESET WEIGHT PRUNING

An algorithm from Baykal et al. (2019b) for pruning the weights of a single perceptron is presented in Algorithm 1; it involves sampling from a distribution over weights, where the probability of being sampled roughly measures the weight's importance to maintaining the output. The algorithm keeps the selected weights (with some re-weighting) and prunes the rest. The resulting pruned perceptron is *guaranteed* to output values that are arbitrarily close to the original perceptron if a sufficiently large number of samples is taken. The relationship between the closeness of the approximation and number of samples required is given by Theorem A.1. Further details and required assumptions are provided in Appendix 3.1. The method can be used to sparsify a whole layer by sparsifying each neuron individually, and bounds for pruning multi-layer neural networks can be constructed using a "layer-by-layer" approach, which is discussed in Section 5.2.

**Deterministic Weight Pruning** Baykal et al. (2019b) also present a deterministic version of their algorithm, which selects the weights with the lowest importances for pruning. This alternative algorithm is presented in Algorithm 2 and the corresponding guaranteed approximation error is presented in Theorem A.2, the proof of which is based on the probabalistic proof of Theorem A.1.

### 3.2 CORESET NEURON PRUNING

Whole neurons can be pruned rather than individual weights. In this case, importances are assigned to each neuron, creating a probability distribution that can be sampled from. A method from Mussay et al. (2020) for pruning feed-forward neural networks with a single hidden layer is shown in Algorithm 3. This algorithm is *data-independent*: it ignores the training data and instead only assumes that the inputs have a bounded L2 norm. Neuron importance is determined by the L2 norm of the parameters of each neuron and the magnitude of the weights assigned to them in the next layer.

Similar to weight pruning, the approximation error can be bounded above depending on the number of samples taken from the distribution over neurons. The relationship is expressed in Theorem A.4. Notice that unlike the *multiplicative* approximation bound provided for weight pruning, this bound is an *additive* error bound, meaning that the guaranteed error term is constant, regardless of the scale of the "true" value. More details about this method are provided in Appendix A.3.

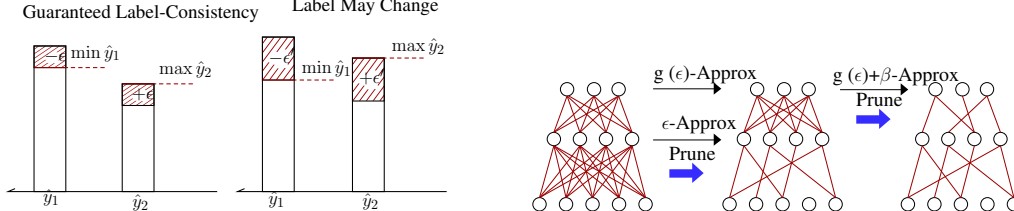

Figure 1: (Left) An illustration of how an additive approximation bound can be used to guarantee label consistency. If each pre-softmax logit can only change by $\epsilon$, then it is impossible for the second-largest logit to surpass the largest logit under pertubation. However, the label may change if each logit can change by $\epsilon'$, which is more than half the distance between them. (Right) In the layer-by-layer approach, we start by pruning the "first" layer (which is closest to the input) with some approximation guarantee $\epsilon$ on the first layer hidden activations. This epsilon error can be "bubbled up" to the output for an implied approximation guarantee on the output $g(\epsilon)$. We proceed to prune the second hidden layer given the pruned first layer activations as input, again noting the guaranteed approximation error $\beta$ and bubbling up to the output. This is repeated for each layer, accumulating approximation error for the deviations in each hidden layer.

### 3.3 Epsilon-First and Sparsity-First Pruning

Pruning using randomized coreset methods involves sampling from a distribution over weights or neurons. The number of samples taken can be determined in one of two ways, depending on whether there is a strict requirement on approximation error or on sparsity:

**Epsilon-first**: The number of samples is determined by the desired epsilon guarantee on the output. Samples are taken with replacement, and sparsity is determined by the number of unique samples.

**Sparsity-first**: Sample from the network until the number of unique samples equals the desired sparsity. Work backwards to infer the epsilon guarantee given that number of samples.

Each approach has its drawbacks. While epsilon-first pruning guarantees a certain accuracy after pruning, it cannot guarantee the sparsity achieved since sampling is done with replacement. Sparsity-first pruning, on the other hand, is not guaranteed to terminate. For example, in the "easy case" experiments in the next section, it will be impossible for the algorithm to achieve 50% sparsity since less than 50% of elements have non-zero probability of being sampled, which will cause the algorithm to run forever.[1]

We will use both approaches in this work, preferring sparsity-first pruning when we must achieve a specific sparsity and preferring epsilon-first pruning when we are familiar with what a "reasonable" amount of approximation error looks like, which we discuss in the next section.

## 4 Evaluating Approximation Guarantees

### 4.1 Accuracy Guarantees

One obstacle to evaluating approximation guarantees is that it is difficult to determine what constitutes a reasonable amount of approximation error. Our solution is to use approximation bounds to predict how many development set labels are guaranteed to remain correct and how many might flip to an incorrect label during pruning. This provides a more intuitive sense of the implications of the approximation bound.

In order to convert an approximation bound into an accuracy guarantee, we must determine whether the output error is large enough to change the prediction of the neural network. Note that the predicted label of a neural network which performs classification is simply the index of the largest

---

[1]The deterministic weight pruning algorithm can also be implemented in a sparsity-first or epsilon-first way. In both, we prune the weights with the lowest sensitivities first. The stopping condition can depend either on the desired sparsity or the desired epsilon approximation guarantee, which we call sparsity-first and epsilon-first pruning, respectively. Unlike the randomized methods, the deterministic method of pruning does not have the associated drawbacks.

pre-softmax logit.[2] Whether the maximum logit will stay the maximum under some perturbation will depend on the distance between the maximum output logit and the second-largest output logit, which varies for each example in the development dataset. This is illustrated in Figure 1 and made precise in the following lemma:

**Lemma 4.1** *Let $f : \mathcal{X} \to \mathcal{Y}$ be a neural network and $x \in \mathcal{X}$ be an arbitrary input. Let $j$ denote the index of the largest pre-softmax logit, $j = \arg \max_i f(x)_i$. Further, let $j'$ denote the second largest pre-softmax logit $f(x)_{j'} \geq f(x)_k, k \neq j$. Finally, let $\hat{f}$ be an approximation of $f$, with guaranteed element-wise approximation error bound $\epsilon$. If the guarantee is additive, then*

$$\epsilon \leq \frac{f(x)_j - f(x)_{j'}}{2} \implies \arg \max_i f(x)_i = \arg \max_i \hat{f}(x)_i$$

*If the bound is multiplicative, then*

$$\epsilon \leq \frac{f(x)_j - f(x)_{j'}}{|f(x)_j| + |f(x)_{j'}|} \implies \arg \max_i f(x)_i = \arg \max_i \hat{f}(x)_i$$

This relationship allows us to determine the epsilon approximation bound that corresponds to some accuracy guarantee after pruning. For example, suppose we want to guarantee 95% of correct labels will stay correct. We would compute the pre-softmax logits of the original neural network on the development set, calculating the epsilon bound required to maintain each correct label. We would then select a final epsilon value which is smaller than 95% of the values computed and prune to achieve that bound.

## 4.2 EASY, MODERATE, AND DIFFICULT CASES

To begin our evaluation, we start with the simplest circumstances: we prune only the final layer of LeNet300-100, a small, two-layer feed-forward neural network trained to solve MNIST. We vary the training procedure to make it easier or harder to prune the network while maintaining low approximation error. More details about each of these cases can be found in Appendix B.

**Pre-pruned Neural Networks** In the easiest case, we pre-prune the neural network to a high sparsity (95% for weight pruning and 90% for neuron pruning) using non-coreset-based heuristic methods and then re-train the network to re-gain all lost accuracy. We then re-initialize the pruned connections to zero. The resulting network is *known to be sparse* and highly prunable while incurring 0 approximation error. We should hope that coreset-based pruning methods will easily prune to the known sparsity while guaranteeing low amounts of approximation error.

**Some Regularization** In the more realistic moderate case, we apply mild L1 regularization (aka Lasso) during the training of the neural network. We set the L1 regularizing penalty to 0.001, which is the highest setting we could use which did not decrease development accuracy.[3]

**No Regularization** In the hardest case, we do not regularize the network. We expect this model to be the least amenable to approximation, and expect the coreset-based methods to appropriately predict that the network cannot be pruned. All models are trained to 98% development accuracy.

## 4.3 REPORT CARD: EPSILON-FIRST PRUNING

Figure 2 shows the results for epsilon-first pruning the final layer of LeNet-300-100 under various training circumstances, guaranteeing that 50% of the development set labels that are correct will remain correct. The full results of these experiments, evaluated at different percentages of accuracy guarantees can be found in Figure 6. As expected, all methods easily achieve high amounts of sparsity in the pre-pruned "easy" case. In the moderate case, less sparsity can be achieved, and in

---

[2]$\arg \max_i \left[ \text{softmax}(f(x))_i \right] = \arg \max_i f(x)_i$, where $f(x) \in \mathcal{Y}$ are the pre-softmax outputs.

[3]For neuron pruning, we might suspect that dropout would be a better choice than L1 regularization to encourage prunability. However, this was not the case in our experiments. We discuss possible explanations and detail our experiments in Appendix B.2.

| Model Type | Prune Type | Case | Required $\epsilon$ Bound | Observed Err | Acc | Layer Size | Samples | Sparsity |
|---|---|---|---|---|---|---|---|---|
| LeNet | Wt Rnd | Easy | 0.46 | 0.04±0.18 | 0.98 | 1K | 271K | 94.06 |
|  |  | Mod | 0.46 | 0.02±0.11 | 0.98 | 1K | 1M | 19.90 |
|  |  | Diff | 0.49 | 0.03±0.18 | 0.98 | 1K | 1M | 0.00 |
|  | Wt Det | Easy | 0.46 | 0.00±0.00 | 0.98 | 1K | 60 | 94.06 |
|  |  | Mod | 0.46 | 0.01±0.04 | 0.98 | 1K | 657 | 34.95 |
|  |  | Diff | 0.49 | 0.01±0.04 | 0.98 | 1K | 927 | 8.22 |
|  | Neuron | Easy | 5.69 | 0.06±0.05 | 0.98 | 100 | 62K | 90.00 |
|  |  | Mod | 3.30 | 0.04±0.03 | 0.98 | 100 | 97K | 16.00 |
|  |  | Diff | 5.20 | 0.13±0.09 | 0.98 | 100 | 114K | 0.00 |
| VGG-16 | Wt Rnd | Easy | 0.31 | 0.01±0.02 | 0.93 | 10K | 88K | 98.90 |
|  |  | Mod | 0.52 | 0.04±0.11 | 0.92 | 10K | 79K | 91.19 |
|  |  | Diff | 0.58 | 0.04±0.22 | 0.92 | 10K | 1.1M | 2.84 |
|  | Neuron | Easy | 8.45 | 0.03±0.02 | 0.93 | 1K | 613K | 99.00 |
|  |  | Mod | 6.63 | 0.01±0.03 | 0.92 | 1K | 501K | 95.00 |
|  |  | Diff | 8.17 | 0.00±0.00 | 0.92 | 1K | 5.5M | 0.00 |

Figure 2: Above are the results for epsilon-first pruning the final layer of LeNet-300-100 and VGG-16 under various training circumstances (Section 4.2), guaranteeing that 50% of the development set labels that are correct will remain correct. We show the calculated epsilon bounds (multiplicative or additive) required to guarantee 50% accuracy (Section 4.1) along with the actual observed approximation error, average and standard deviation. We also show the layer size (number of neurons or number of weights) as well as the number of samples required by the theorem to meet the epsilon bound. Note that the sparsity may be different than the number of samples required, since sampling is done with replacement.

the difficult case almost no pruning can be done. However, we note some practical issues with the performance of these methods:

**High Sample Complexity** We can see (highlighted in red) that the number of samples required during pruning is unnecessarily high. Even in the easy case, in which we know the network is already sparse, the number of samples required is three orders of magnitude larger than the number of elements to sample from. We are still able to achieve some sparsity in the easy and moderate cases, however, since the probability distribution over weights and neurons is highly skewed. High sample complexity ($< 1M$ samples) can easily be handled by consumer hardware, but may grow impractical as the layers grow larger.

**Loose Bounds** We also note that the observed approximation error is much lower than the provided bound (highlighted in orange). We guaranteed only 50% of the accuracy should be maintained, but in all cases 100% of the accuracy was maintained. The looseness of the bounds implies that much higher sparsities could be reached while still achieving the desired accuracy guarantee. Indeed, Figure 3 shows that even the "difficult" case neural network can be pruned up to 90% sparsity before reaching 50% accuracy, whereas our best method only reached 8% sparsity. For neuron pruning, we might suppose that the looseness of bounds can be explained by laxness of the data-independence assumption. We explore this idea in Appendix C.1, but fail to conclude that lax data assumptions are a sufficient cause of loose bounds.

The deterministic version of weight pruning, while still providing significantly loose bounds, sidesteps the sample complexity issue by not requiring any random sampling (highlighted in green). In our opinion, this is the most promising future for these methods, which may leverage the benefits of probabalistic proof techniques while minimizing their drawbacks via de-randomization. However, *none of these methods provide approximation bounds which are fine-grained enough to predict the sparsity/accuracy trade-off,* which we expect to become a theme of future work.

### 4.3.1 SPARSITY-FIRST PRUNING

The high sample complexity of epsilon-first pruning implies that sparsity-first pruning with randomized methods will lead to incredibly vacuous approximation bounds. This is because the number of samples taken is usually less than the number of elements to be sampled from. And indeed, Figure 8 shows that pruning a "moderate case" network at weight sparsities greater 35% lead to bounds that cannot guarantee any accuracy at all. Pruning at less that this sparsity causes the algorithm to

run forever. Sparsity-first pruning with deterministic weight pruning is easier to work with, but still under-estimates the achieved accuracy after pruning due to loose bounds.

### 4.3.2 WHAT DRIVES SAMPLE COMPLEXITY?

Two main components in each theorem drive sample complexity: the tightness of the epsilon bound and the "total sensitivity" of the layer. Sensitivity is a measure of *importance*, so if all neurons or weights are equally important then we will require a large number of samples to achieve a good approximation. If only a few weights are important, then total sensitivity will be small and we will require fewer samples.

In the case of weight pruning, sample complexity scales linearly with total sensitivity, and the slope of the line is determined by the required epsilon bound and the probability of failure.[4] For LeNet, layers tend to have a relatively low total sensitivity (only 5% of the possible maximum). Even at the minimum total sensitivity, however, Theorem A.1 would require sampling 1.5x the number of weights for each perceptron. This seems to imply that sample complexity's quadratic dependence on epsilon is too restrictive for neural networks of the size typically deployed in practice, and that future work should focus on relaxing that dependence.

For neuron pruning in Theorem A.4, the sensitivity of each neuron is a function of the norm of the inputs and the norm of the weights of each neuron. This leads us to Corollary C.2, which says that we can *arbitrarily increase or decrease the sample complexity* of any accuracy guarantee by scaling the appropriate weight matrices by a constant factor. We prove this using the *positive homogeneity* property of ReLU networks (Du et al., 2018a)[5], and verify it empirically in Appendix C.2. However, investigating the full implications of this observation are out of the scope of this paper.[6]

### 4.3.3 SCALING UP TO LARGER NETWORKS

We might expect that results would be more compelling on larger neural networks trained on more realistic data, since coreset algorithms were originally applied to big-data streaming applications in which millions of elements are sampled from. To test this, we repeat the experiments above on a VGG-16 model trained to 92% development accuracy on CIFAR-10, which has 10x the number of neurons in the previous experiments. We find that these models are more prunable via heuristic methods than LeNet; we can achieve 99% weight and neuron sparsity without decreasing accuracy.

The results for coreset-based methods, however, are qualitatively similar to earlier experiments (Figure 2 and Appendix B.3). While the number of neurons has increased by 10x, so too has the number of samples required by deterministic weight pruning and randomized neuron pruning. Bounds are similarly loose for all methods. The relative increase in weight pruning sample complexity, however, is much less than 10x (highlighted in blue), which we believe reflects the increased prunability of these models. This may imply that coreset-based methods may give non-vacuous sampling complexities for very large and very sparse layers. However, the scale required (10k+ hidden units) does not seem to overlap with sizes typically used in practice.

## 5 FINE-TUNING DOES NOT APPROXIMATE

In the sections above, we have only evaluated networks immediately after pruning, without any fine-tuning allowed. However, fine-tuning is an integral step in the pruning process. This is illustrated in Figure 3, in which we prune a "difficult case" neural network to 95% weight sparsity using a sparsity-first coreset method. Accuracy drops to below 40% after pruning, but recovers to 98% after fine-tuning. We call this discrepancy between accuracy before and after fine-tuning the "fine-tuning gap." While approximation bounds may improve accuracy prediction after pruning by tightening bounds, we will see in the following sections that they have no hope of predicting the size of the

---

[4]We can determine the number of samples required by solving for $m$ in terms of $\epsilon$, see Corollary 5 of Baykal et al. (2019b).

[5]In ReLU networks, positive homogeneity means that multiplying a weight matrix by a constant multiplies the pre-softmax output by the same factor. This "decreases the temperature" of the softmax distribution.

[6]This scaling trick cannot be applied to Theorem A.1 for weight pruning, since the sensitivities and epsilon bounds are scale-invariant.

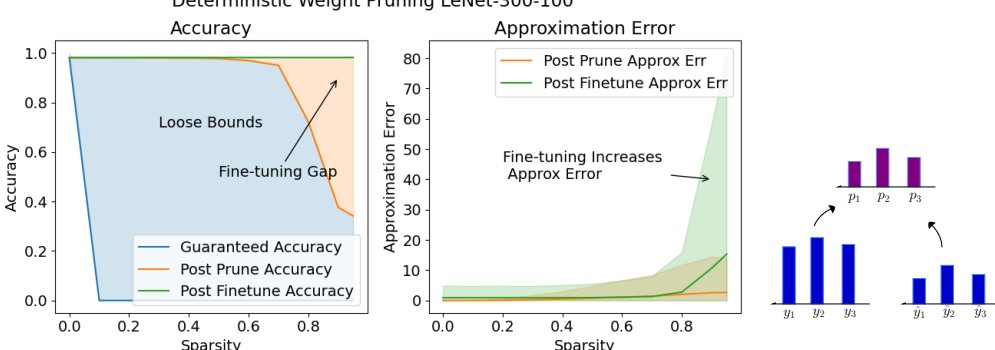

Figure 3: (Left) The results of pruning the final layer of "difficult case" LeNet-300-100 with sparsity-first deterministic weight pruning. The provided accuracy guarantees vastly under-estimate the accuracy after pruning, which can be attributed to loose bounds. Fine-tuning increases accuracy after pruning by more than 50%, which we call the fine-tuning gap. (Middle) Fine-tuning increases approximation error in the pre-softmax logits rather than decrease it. This implies that approximation bounds would not be a good predictor of the size of the fine-tuning gap. (Right) An illustration of softmax translation invariance, where $\hat{y} = y - C$. Both pre-softmax activations lead to exactly the same probability distribution over classes, since the output of softmax depends only on the difference between logits, not their magnitude.

fine-tuning gap while in their current form. This is because fine-tuning allows us to recover accuracy while simultaneously maintaining or increasing the approximation error measured by these bounds.

### 5.1 SOFTMAX TRANSLATION INVARIANCE

First, the methods described above measure approximation error before the softmax layer is applied. However, the softmax layer is *translation invariant*, which means that adding a constant $C$ to each pre-softmax logit will not change the output distribution. This is illustrated in Figure 3 (Right). This means gradient descent during fine-tuning will not have any incentive to reduce pre-softmax approximation error, but instead will be driven to adjust each logit locally with respect to the other logits in the output. This is true both when the pruned network is fine-tuned on the training data and when it is trained using knowledge distillation (Hinton et al., 2015).

This can be observed in Figure 3, which also shows the approximation error before and after fine-tuning on the training data for a "difficult case" network pruned using deterministic sparsity-first weight pruning. Although accuracy recovers from 30% to 98%, the pre-softmax approximation error increases or stays the same after fine-tuning. Given the fragile relationship between accuracy and pre-softmax approximation error, we can expect methods which rely on such approximation bounds to be poor predictors of performance after fine-tuning.

### 5.2 LAYER-BY-LAYER BOUNDS

A "layer-by-layer" approach may be used to prune multi-layer neural networks with approximation guarantees, as illustrated in Figure 1. While layer-by-layer construction of approximation bounds is conceptually simpler and amenable to proof than working with the network as a whole, it requires that the pruned neural network approximate the original dense network at each hidden layer. If the approximation error at any intermediate layer is large, it "bubbles up" and greatly increases the approximation error at the output.

In practice, however, fine-tuning allows for later layers to compensate for large amounts of approximation error in layers close to the input. Consider the table below: we prune the first layer of a "difficult case" neural network to a high sparsity and observe the resulting approximation error in both the first hidden layer and the output. We then fine-tune the model to approximate the original network's pre-softmax output activations, as in patient knowledge distillation (Sun et al., 2019).[7]

[7]Although fine-tuning has no incentive to approximate the original dense network's pre-softmax activations, we may artificially induce this behavior by explicitly minimizing the pre-softmax approximation error during

| Model | Prune Type | Prune | Prune Acc | Output Error | Hidden Error | Post-KD Acc | Output Error | Hidden Error |
|-------|-----------|-------|-----------|--------------|--------------|-------------|--------------|--------------|
| LeNet | Neuron | 0.90 | 0.4809 | 6.78±5.68 | 1.22±1.08 | 0.9513 | 2.95±2.83 | 1.72±1.44 |
|       | Wt Det | 0.95 | 0.773 | 1.37±5.11 | 1.90±4.89 | 0.9762 | 0.71±0.84 | 1.96±5.10 |
| VGG-16 | Wt Det | 0.99 | 0.2474 | 1.51±5.44 | 2.05±5.57 | 0.92 | 0.87±0.39 | 2.04±5.62 |

This process indeed reduces the pre-softmax approximation error of the pruned network and recovers most of the development accuracy. However, the approximation error in the hidden layer remains the same or increases (highlighted in red). This means that any layer-by-layer approximation bound will be "bottlenecked" by this hidden error, even though the pruned neural network has low approximation error when considered as a whole.

## 5.3 PRUNABLE $\neq$ APPROXIMABLE

The results of the previous sections highlights a subtle distinction between layers that are easy to approximate and layers which are prunable. Pruning a layer may introduce a large amount of approximation error. If the layer is near the output, then the network may still output the correct labels by exploiting the translation invariance of softmax. If the layer is near the input, then later layers may adapt to the approximation error by adjusting their weights. In the next section, we discuss the implications of this distinction on future work on pruning with approximation bounds.

## 5.4 WHEN ARE APPROXIMATION BOUNDS USEFUL?

**Pruning Efficacy** Given the subtle distinction between prunable and approximable networks, it is not clear whether pruning methods which directly attempt to approximate the original neural network should be able to prune more effectively in terms of the development accuracy / sparsity trade-off. While Mussay et al. (2020) do not compare to any heuristic methods after re-training, Baykal et al. (2019b) find only small improvements (1-2%) in development accuracy over magnitude weight pruning at the highest levels of sparsity.[8]

It is out of the scope of this paper to evaluate these claims, since we are mainly concerned with the predictive power of these methods and the guarantees they provide. However, it is possible that minimizing approximation error improves the ability of gradient descent to find "other" global optima that are nearby. This would make sense in light of mode-connectivity (Garipov et al., 2018), which observes that neural networks may have many global optima connected by simple linear paths. However, a more rigorous analysis is required to provide a motivation and quantitative prediction of the benefits of approximation.[9]

**Optimal Layer Sparsity** Another claimed benefit of good approximation bounds is the ability to correctly predict the appropriate sparsity for each hidden layer or neuron, without having to empirically test every possible sparsity combination. This is done in Baykal et al. (2019b) using a convex optimzation method to minimize the sum of the errors in each layer. However, we have observed that different layers can tolerate different amounts of approximation error due to the ability of later layers to compensate during fine-tuning. We believe it is unlikely that approximation error theory by itself will allow for useful predictions about optimal sparsity; optimizability of sparse networks dur-

---

gradient descent, which is similar in spirit to knowledge distillation (Hinton et al., 2015). However, training is non-trivial due to exploding gradients caused by the large difference in scale between pruned and original activations. Normalizing the activations by their L2-norm before calculating the loss can mitigate this, but allows for inexact approximation during fine-tuning.

[8]It should be noted, however, that there is a crisis in evaluating neural pruning methods, since the results are notoriously difficult to replicate (Blalock et al., 2020) (Gale et al., 2019). Any claims that one pruning method is "better" than another should be carefully evaluated. We should also take into consideration of the complexity required to execute the methods. In terms of the ratio of performance to simplicity, we believe magnitude pruning and re-training remains un-paralleled.

[9]This might benefit from an extension of (Du et al., 2018b), which predicts the circumstances in which gradient descent will find a global optimum, to the case of sparse training. An analysis might also benefit from the observation that re-winding the weights to initialization before fine-tuning can sometimes lead to better performance (Renda et al., 2020).

ing fine-tuning must be taken into account. It may also be the case that deviating from the original network may be a desirable characteristic, as the new optimum may be more generalizable.

**Generalization** Pruning approximation bounds were made recently popular by their application to proofs of generalization, and that is likely where they will remain useful. However, proofs of generalization which employ loose approximation bounds will likely result in loose generalization bounds, and so care should be taken by the consumer of these theorems.

## 6 CONCLUSION

There is still work to be done in order to make neural pruning approximation bounds both practical and correct. Current methods tend to over-estimate the approximation error that will be observed in practice, and randomized methods require an impractical number of samples to be taken during pruning, limiting the sparsity that can be achieved while guaranteeing fidelity. There are a number of avenues to improve these short-comings, including focusing on derandomization, taking into account the positive homogeneity of neural networks, and only applying coreset methods to networks of an appropriate scale.

However, while these improvements may tighten approximation bounds, they will still not be practically predictive of performance unless they take into account the invariance of the softmax layer and are accompanied by an analysis of the fine-tuning process. Both of these facets of neural network training allow pruned networks to accrue high amounts of approximation error in hidden layers while maintaining label-consistency with the original dense networks. Predicting the outcome of pruning and re-training, which is an indispensible step in all effective pruning methods, will require a more sophisticated analysis of the optimization of sparse neural networks via gradient descent.

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

## A  CORESET METHOD DETAILS

### A.1  NOTATION

We are interested in pruning a feed-forward neural network with $L \in \mathbb{N}_+$ layers which defines a mapping $f : \mathcal{X} \to \mathcal{Y}$ for an input $x \in \mathcal{X} \subset \mathbb{R}^d$ to an output $y \in \mathcal{Y} \subset \mathbb{R}^k$. Borrowing the notation of Baykal et al. (2019a), let $\eta^{(\ell)} \in \mathbb{N}_+$ denote the number of neurons in layer $\ell \in [L]$, where $[L] = \{1, \dots, L\}$ denotes the index set. Note $\eta^{(1)} = d$ and $\eta^{(L)} = k$. Further, let $\eta = \sum_{\ell=1}^{L} \eta^{(\ell)}$.

---

**Algorithm 1:** Randomized Coreset-based Weight Pruning of Single Neuron

---

**Input:** $\mathcal{I}$: indices of non-negative parameters; $w$: neuron parameters to sparsify; $m$: number of parameters to sample; $S \subset P$: batch of validation points; $K$: constant from Asm. A.3

**Output:** $\hat{w}$: sparse weight tensor

1   $s_j \leftarrow \max_{x \in S} \frac{w_j x_j}{\sum_{k \in \mathcal{I}} w_k x_k}; \forall j \in \mathcal{I}$ {Compute parameter importances}

2   $q_j = s_j / \sum_{k \in \mathcal{I}}; \forall j \in \mathcal{I}$

3   $\mathbf{K} \sim \text{MULTINOMIAL}(q, m);$

4   **for** $j \in \mathcal{I}$ **do**

5       $\hat{w}_j \leftarrow \frac{\mathbf{K}_j w_j}{m q_j}$ { Re-weight entries to ensure unbiasedness of estimator }

6   **end**

7   return $\hat{w}$;

---

**Algorithm 2:** Deterministic Coreset-based Weight Pruning of Single Neuron

---

**Input:** $\mathcal{I}$: indices of non-negative parameters; $w$: neuron parameters to sparsify; $m$: number of parameters to sample; $S \subset P$: batch of validation points; $K$: constant from Asm. A.3

**Output:** $\hat{w}$: sparse weight tensor

1   $s_j \leftarrow \max_{x \in S} \frac{w_j x_j}{\sum_{k \in \mathcal{I}} w_k x_k}; \forall j \in \mathcal{I}$ {Compute parameter importances}

2   $\mathcal{I}_m \leftarrow$ indices of the $m$ largest values of $(s_j)_{j \in \mathcal{I}}$ where ties are broken arbitrarily;

3   **for** $j \in \mathcal{I}_m$ **do**

4       $\hat{w}_j \leftarrow w_j$

5   **end**

6   return $\hat{w}$;

---

**Algorithm 3:** Coreset-based Neuron Pruning of Single Hidden Layer

---

**Input:** $W^{(1)} \in \mathbb{R}^{\eta^{(2)} \times d}$: weights of hidden layer; $W^{(2)} \in \mathbb{R}^{k \times \eta^{(2)}}$ weights of output layer; $m$: number of neurons to sample; $\beta > 0$: an upper bound on the L2 norm of the input values.

**Output:** $\hat{W}^{(1)}$: pruned hidden layer; $\hat{W}^{(2)}$: new weights for output layer.

1   $q_j \leftarrow \frac{\max_{i \in [k]} w_{ij}^{(2)} \phi(\beta || w_j^{(1)} ||)}{\sum_{j' \in [\eta^{(2)}]} \max_{i \in [k]} w_{ij'}^{(2)} \phi(\beta || w_{j'}^{(1)} ||)}; \forall j \in [\eta^{(2)}]$ {Compute neuron importances}

2   $\mathbf{K} \sim \text{MULTINOMIAL}(q, m);$

3   **for** $j \in [\eta^{(2)}]$ **do**

4     **if** $\mathbf{K}_j \neq 0$ **then**

5         $\hat{w}_j^{(1)} \leftarrow w_j^{(1)}$

6         $\forall i \in [k] : \hat{w}_{ij}^{(2)} \leftarrow \frac{\mathbf{K}_j w_{ij}^{(2)}}{m q_j}$

7     **end**

8     **else**

9         $\hat{w}_j^{(1)} \leftarrow \vec{0}$

10         $\forall i \in [k] : \hat{w}_{ij}^{(2)} \leftarrow 0$

11     **end**

12   **end**

13   return $\hat{W}^{(1)}, \hat{W}^{(2)}$;

---

Figure 4: (Top, Middle) Algorithms from Baykal et al. (2019b) for pruning a single perceptron with positive weights via a data-dependent coreset method. (Bottom) Algorithm from Mussay et al. (2020) for pruning a single hidden-layer feed-forward neural network data-independent coreset method.

For each layer $\ell \in \{2, \ldots, L\}$, let $W^{(\ell)} \in \mathbb{R}^{\eta^{(\ell)} \times \eta^{\ell-1}}$, be the weight matrix for layer $\ell$ with entries denoted by $w_{ij}^{(\ell)}$ and rows denoted by $w_i^{(\ell)} \in \mathbb{R}^{1 \times \eta^{\ell-1}}$. For notational simplicity, we assume that the bias is embedded in the weight matrix. Finally, let $\rho^\ell = \mathrm{nnz}(W^{(\ell)})$ be the number of parameters in the matrix $W^{(\ell)}$ and $\rho^* = \max_{\ell \in [L]} \rho^\ell$. We consider the activation to be the Rectified Linear Unit (ReLU) function, i.e., $\phi(\cdot) = \max\{\cdot, 0\}$. If the network is used for a classification task, then the predicted label is $\arg\max_i \left[\mathrm{softmax}(f(x))_i\right] = \arg\max_i f(x)_i$.

## A.2 WEIGHT PRUNING

When sparsifying weights with coreset-based methods, we can guarantee that the output of the pruned perceptron is arbitrarily close to the original if a sufficient number of samples is taken. A bound on the approximation error given a certain number of samples is presented in the following theorem:

**Theorem A.1** *(Baykal et al., 2019b) Let $K, K' \in \mathbb{R}$ be constants that can be determined from Assumption A.3. Let $\mathcal{I}$ indicate the indices of the non-negative weights of the perceptron. For $\delta \in (0,1)$, invoking Alg 1 with $m \in \mathbb{N}$ satisfying $m > \tilde{S}$, and a set $S \subset \mathcal{X}$ composed of $\lceil K' \log(2\rho/\delta) \rceil$ i.i.d. points drawn from $\mathcal{D}$ generates a perceptron $\hat{w}$ such that $\mathrm{nnz}(\hat{w}) \leq m$ and for some $x \sim \mathcal{D}$,*

$$Pr(|\hat{z}(x) - z(x)| \geq \epsilon_m z(x)) \leq \delta$$

*where $z(x) = \sum_{k \in \mathcal{I}} w_k x_k$, $\hat{z}(x) = \sum_{k \in \mathcal{I}} \hat{w}_k x_k$,*

$$\epsilon_m = \left( \sqrt{\frac{\tilde{S}}{m}\left(\frac{\tilde{S}}{m} + 6\right)} + \frac{\tilde{S}}{m} \right) \in (0, 1)$$

*$\tilde{S} = SK \log(4/\delta)$, and $S = \sum_{j \in \mathcal{I}} s_j$*

The theorem provides a probabilistic guarantee, meaning our confidence that the guarantee will hold can be adjusted via a parameter $\delta$. In our experiments, we use $\delta = 0.5$ unless otherwise specified. Note that this result holds only when the weights of the neural network are positive. A generalization to all weights is given by Lemma 6 of Baykal et al. (2019b). This involves sparsifying the positive and negative weights separately and then combining the results, which is how we implement this method for our experiments.

Baykal et al. (2019b) also present a deterministic version of their algorithm, which prunes the lowest-sensitivity weights first. The approximation error can also be bounded in this case using the following theorem:

**Theorem A.2** *(Baykal et al., 2019b) In the context of Theorem A.1, invoking Algorithm 2 generates a perceptron $\hat{w}$ such that for some $x \sim \mathcal{D}$.*

$$Pr(|\hat{z}(x) - z(x)| \geq \epsilon_m z(x)) \leq \frac{\delta}{2}$$

*where $z(x) = \sum_{k \in \mathcal{I}} w_k x_k$, $\hat{z}(x) = \sum_{k \in \mathcal{I}} \hat{w}_k x_k$, and $\epsilon_m = 3K \sum_{j \in (\mathcal{I} \setminus \mathcal{I}_m)} s_j$*

The proof of both weight pruning approximation bounds requires an assumption on the Cumulative Distribution Function (CDF) of the sensitivities of each weight. For our work, we use values of $K' = 3$ and $K = 2$ which are recommended by the authors.

**Assumption A.3** *(Baykal et al., 2019b) Let $g_j(x) = w_j x_j / \sum_{k \in \mathcal{I}} w_k x_k$, for $x \sim \mathcal{D}$. There exist universal constants $K, K' > 0$ such that for all $j \in \mathcal{I}$, the CDF of the random variable $g(x)_j$ for $x \sim \mathcal{D}$, denoted by $F_j(\cdot)$, satisfies $F_j(M_j/K) \leq \exp(-1/K')$, where $M_j = \min\{y \in [0, 1] : F_j(y) = 1\}$.*

### A.3 NEURON PRUNING

Neurons can be pruned rather than weights via Algorithm 3. A bound on the approximation error of the pruned neural network can be guaranteed if a sufficient number of samples is taken. This is expressed in the following theorem:

**Theorem A.4** *Mussay et al. (2020) Let $W^{(1)} \in \mathbb{R}^{\eta^{(2)} \times d}$ be the weights of the hidden layer and $W^{(2)} \in \mathbb{R}^{k \times \eta^{(2)}}$ be the weights of the output layer, and $\beta$ be an upper bound on the L2-norm of $x \in \mathcal{X}$, as in Alg 3. Let*

$$s_j = \max_{i \in [k]} w_{ij}^{(2)} \phi(\beta ||w_j^{(1)}||_2)$$

*be the sensitivities of each neuron for $\forall j \in [\eta^{(2)}]$, and $t = \sum_j s_j$ be the "total sensitivity." Let $c > 1$ be a sufficiently large constant that can be determined from the proof, and*

$$m \geq \frac{ct}{\epsilon^2} \left( d \log t + \log\left(\frac{1}{\delta}\right) \right)$$

*be the sample size. Let $\hat{W}^{(1)}, \hat{W}^{(2)}$ be the output to a call to Algorithm 3. Then, with probability at least $1 - \delta$,*

$$\forall i \in [k], x \in \mathbb{B}_\beta(0) : \left| \sum_{j \in [\eta^{(2)}]} w_{ij}^{(2)} \phi(w_j^{(1)} \cdot x) - \sum_{j \in [\eta^{(2)}]} \hat{w}_{ij}^{(2)} \phi(\hat{w}_j^{(1)} \cdot x) \right| \leq \epsilon$$

To avoid the confusion of introducing new notation, we have re-written this theorem using the notation used in Baykal et al. (2019a). We also do not mention VC-dimension in this theorem, which is needed for the general case. In our case, the VC-dimension of a single perceptron with ReLU activation is simply the input dimension, $d$.

Neuron pruning approximation bounds require knowing the maximum L2 norm of the input vectors. This can be calculated naively; for example, MNIST has 784 input dimensions each within $(0, 1)$. Then the maximum L2 norm is $\sqrt{784} = 28$; we could then bound the L2 norm of hidden activations by examining the eigenvalues of each weight matrix. In practice, however, MNIST has highly sparse inputs and we observe the maximum L2 norm to be 25. In this case, we set $\beta$ to be the lower value and follow a similar empirical approach to estimate the appropriate maximum L2 norm of hidden activations.

## B EXPERIMENTAL DETAILS

### B.1 LENET TRAINING DETAILS

LeNet-300-100 is a simple two-layer feed-forward neural network with 300 hidden units in the first layer and 100 hidden units in the second layer. All LeNet-300-100 models are trained to achieve 98% development accuracy on MNIST. Training is carried out for 10 epochs with a standard SGD optimizer at learning rate 0.1, momentum 0.5, and a training batch size of 100.

**Easy Case** There are two "easy case" LeNet models, one which pre-prunes the neurons of the final hidden layer and one which pre-prunes the weights of the final weight matrix. Weight pruning is done via *magntiude weight thresholding* (Han et al., 2015), which is the most popular form weight pruning.

Neuron pruning is accomplished via *L0 regularization*, which adds a penalty to the loss function of a neural network depending on the number of neurons with non-zero weights (the L0-norm of the neurons). This loss function is computationally intractable to optimize exactly, but in practice can be optimized via a stochastic relaxation (Louizos et al., 2017). In this method, each neuron is multiplied

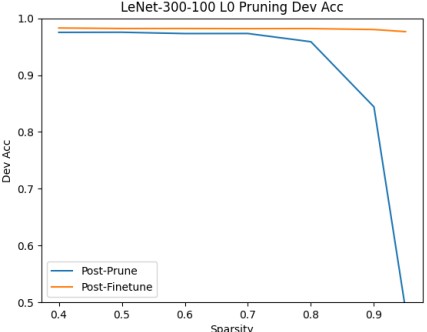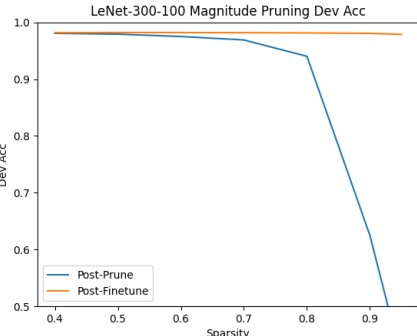

Figure 5: The development accuracy of LeNet-300-100 models after pruning the final layer using heuristic methods and also after fine-tuning. On the left is neuron pruning via L0 regularization, and on the right is weight pruning via magnitude weight pruning.

by a gate variable which probabilistically takes on values in $[0, 1]$. The gates are sampled from a *hard-concrete* distribution (Maddison et al., 2017), which is amenable to the re-parameterization trick (Kingma & Welling, 2014) and therefore to optimization via backpropogation. After training, we prune the neurons whose gates have the highest probability of being "off." Specific values of L0 pruning hyperparameters can be found in our codebase.

Once easy-case networks are pruned, we re-train the sparse networks for another 10 epochs to regain all lost accuracy and then re-initialize the pruned weights to 0. This results in networks that are known to be sparse and prunable with no approximation error. The development accuracies both before and after fine-tuning are shown in Figure 5.

**Moderate Case** In the moderate case of training, we apply mild L1 regularization on the weights. We tried regularization parameters $[1, 0.1, 0.01, 0.001, 0.0001]$, and found that 0.001 was the largest value we could use without impacting development accuracy.

**Difficult Case** In the difficult case, we do not apply any additional regularization during training.

## B.2 TRAINING WITH DROPOUT

While applying L1 regularization improved neuron prunability, applying dropout at rates of 0.2 and 0.5 did not (Figure 7). While at first surprising, this makes sense upon further consideration. Dropout is known to "break co-adaptation" between neurons (Mianjy & Arora, 2019), meaning that any random subset of the neurons should produce approximately the same answer. In turn, this means that no single neuron should have it's weights go to zero; all neurons maintain roughly the same importance.

Because of this, each neuron is assigned roughly a uniform probability of being sampled by our coreset method. And because the sample complexity is higher than the number of neurons in the layer, this results in 0% sparsity being achieved. This is somewhat ironic, since any particular choice of 50% of the neurons should have resulted in decent post-pruning performance.

## B.3 VGG-16 DETAILS AND RESULTS

VGG-16 is a neural network architecture which is composed of several convolutional layers followed by a 3 feed-forward layers with 4096, 4096, and 1000 hidden units respectively. All VGG-16 models are trained to achieve 92% development accuracy on CIFAR-10. Training is carried out for 15 epochs with a standard SGD optimizer at learning rate 0.001, momentum 0.9, and a training batch size of 32. The full results of this section are shown in Figure 10.

"Easy case" VGG-16 networks are pruned using either weight pruning and L0 pruning, similar to the LeNet models. Unlike the LeNet models, the final layer of VGG-16 could be pruned to 99% before accuracy began to decrease after fine-tuning (Figure 9).

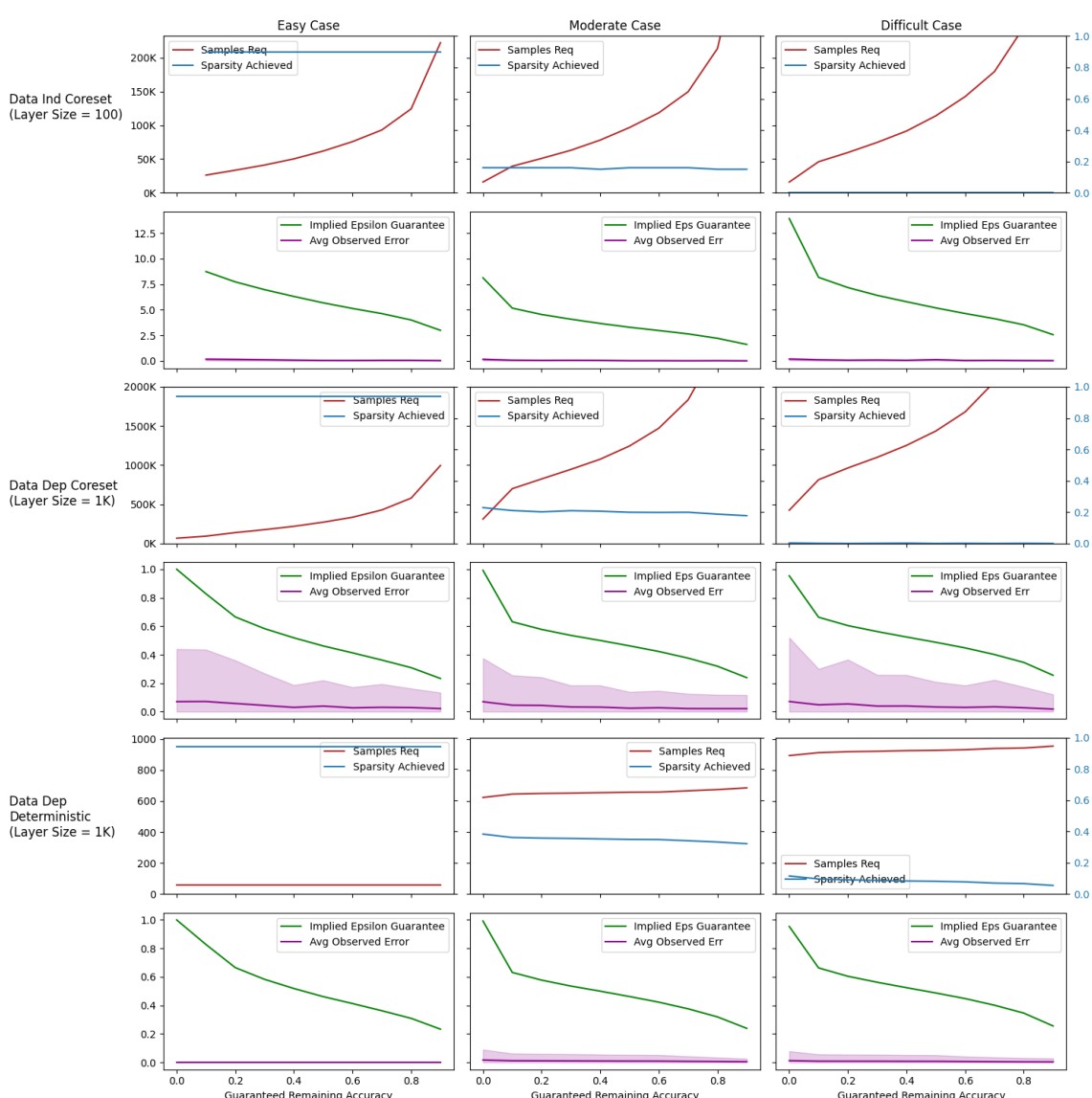

Figure 6: The full results of pruning the final layer of LeNet-300-100 under easy, moderate, and difficult training circumstances (Section 4.2). Shown are the implied epsilon bounds required to reach each desired accuracy guarantee (additive for neuron pruning, multiplicative otherwise), as well as the observed approximation (average and standard deviation). We also show the number of samples required to meet the epsilon bound as well as the sparsity achieved after sampling.

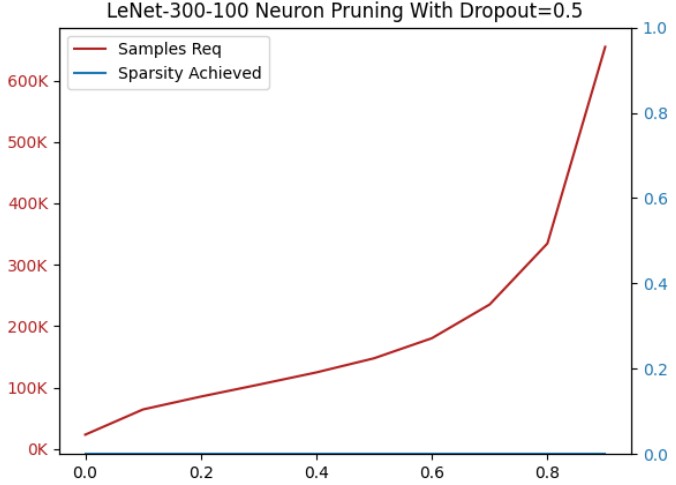

Figure 7: Samples required and sparsity (which is 0) for pruning LeNet-300-100 neurons via a coreset method, when dropout was applied during training at a rate of 0.5.

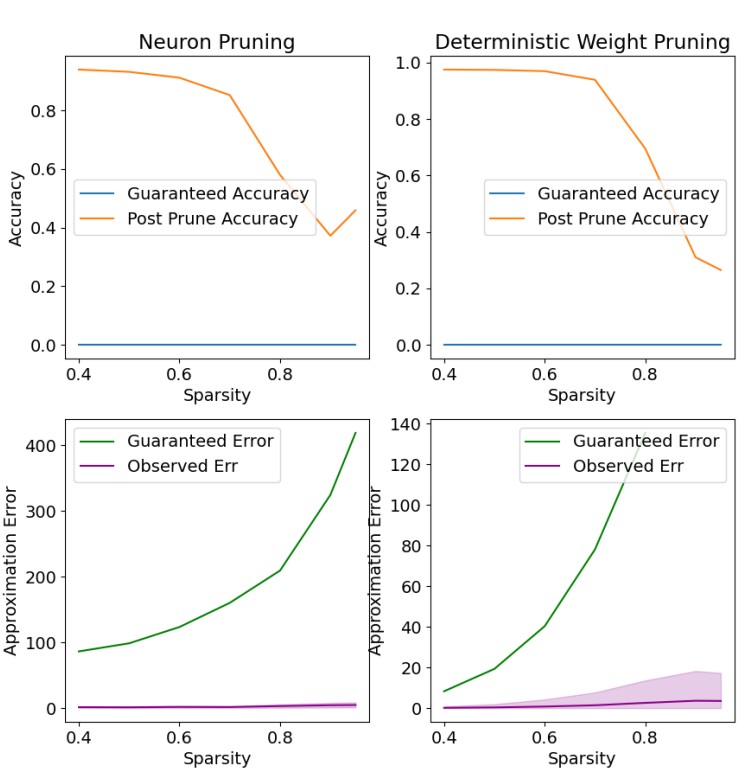

Figure 8: Results for pruning a "moderate case" LeNet-300-100 with sparsity-first methods. Shown are the accuracy and approximation error achieved during pruning, as well as the implied accuracy guarantee and implied approxiation error bound computed from the number of samples taken.

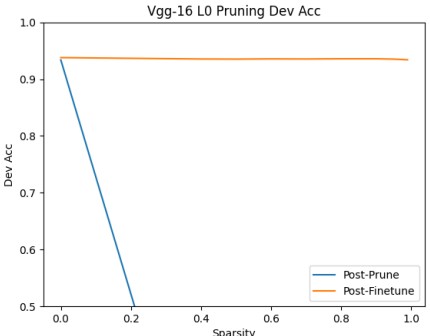 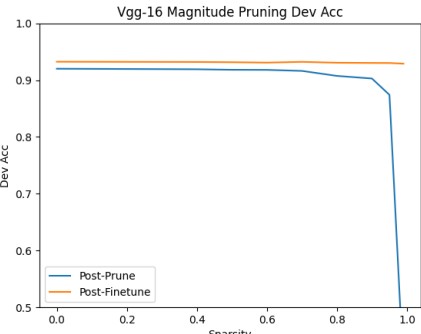

Figure 9: The development accuracy of VGG-16 models after pruning the final layer using heuristic methods and also after fine-tuning. On the left is neuron pruning using L0 regularization, and on the right is weight pruning using magnitude weight pruning.

"Moderate case" VGG-16 models were trained with an L1 regularization term of 0.001, which was determined in a similar manner to that of LeNet-300-100.

## C  FURTHER ANALYSIS OF SAMPLE COMPLEXITY

### C.1  DOES DATA-INDEPENDENCE MEAN LOOSE BOUNDS?

We might assume that the looseness of the neuron pruning bounds can be attributed to overly-lax assumptions about the data distribution (i.e. that the L2 norm of inputs is simply bounded above). To test this hypothesis, we generate random points on the sphere used to bound input vectors and measure the approximation error on these inputs, with the results shown in Figure 11. Randomizing the inputs does increase approximation error, but it does not increase enough to justify the scale of the bounds. We believe this implies that the bounds can be made tighter while maintaining the data-independence assumption.

### C.2  MANIPULATING TOTAL SENSITIVITY

In this section we will demonstrate that the number of samples required by coreset neuron pruning to guarantee label-consistency at any accuracy can be arbitrarily manipulated. This involves manipulating both the total sensitivity of the layer and the distance between pre-softmax logits. First we'll prove the following lemma:

**Lemma C.1** *Suppose $\phi(\cdot) = \max(0, \cdot)$. Then the "total sensitivity" $t$ of a hidden layer, as defined in Theorem A.4 for pruning neurons, can be made arbitrarily small or large by multiplying the weight matrix $W^{(1)}$ by a positive constant $\alpha$. The resulting neural network is label-consistent with the original.*

**Proof :**  The first statement is fairly straight-forward to prove. Let $\hat{W}^{(1)} = \alpha W^{(1)}$ be a modified weight matrix of the first layer, where $\alpha > 0$ is some arbitrary constant. Similarly, let $\hat{t}$ be the resulting total sensitivity of the network which uses $\hat{W}^{(1)}$, rather than $W^{(1)}$. Then

$$\hat{t} = \sum_j \hat{s}_j = \sum_j \max_{i \in [k]} w_{ij}^{(2)} \phi(\beta || \hat{w}_j^{(1)} ||_2)$$

$$= \sum_j \max_{i \in [k]} w_{ij}^{(2)} \phi(\beta || \alpha w_j^{(1)} ||_2) = \alpha \sum_j \max_{i \in [k]} w_{ij}^{(2)} \phi(\beta || w_j^{(1)} ||_2) = \alpha t$$

So scaling $W^{(1)}$ by $\alpha$ also scales the total sensitivity by $\alpha$. The fourth equality holds because $\max(0, \alpha x) = \alpha \max(0, x)$ and $||\alpha w||_2 = \alpha ||w||_2$.

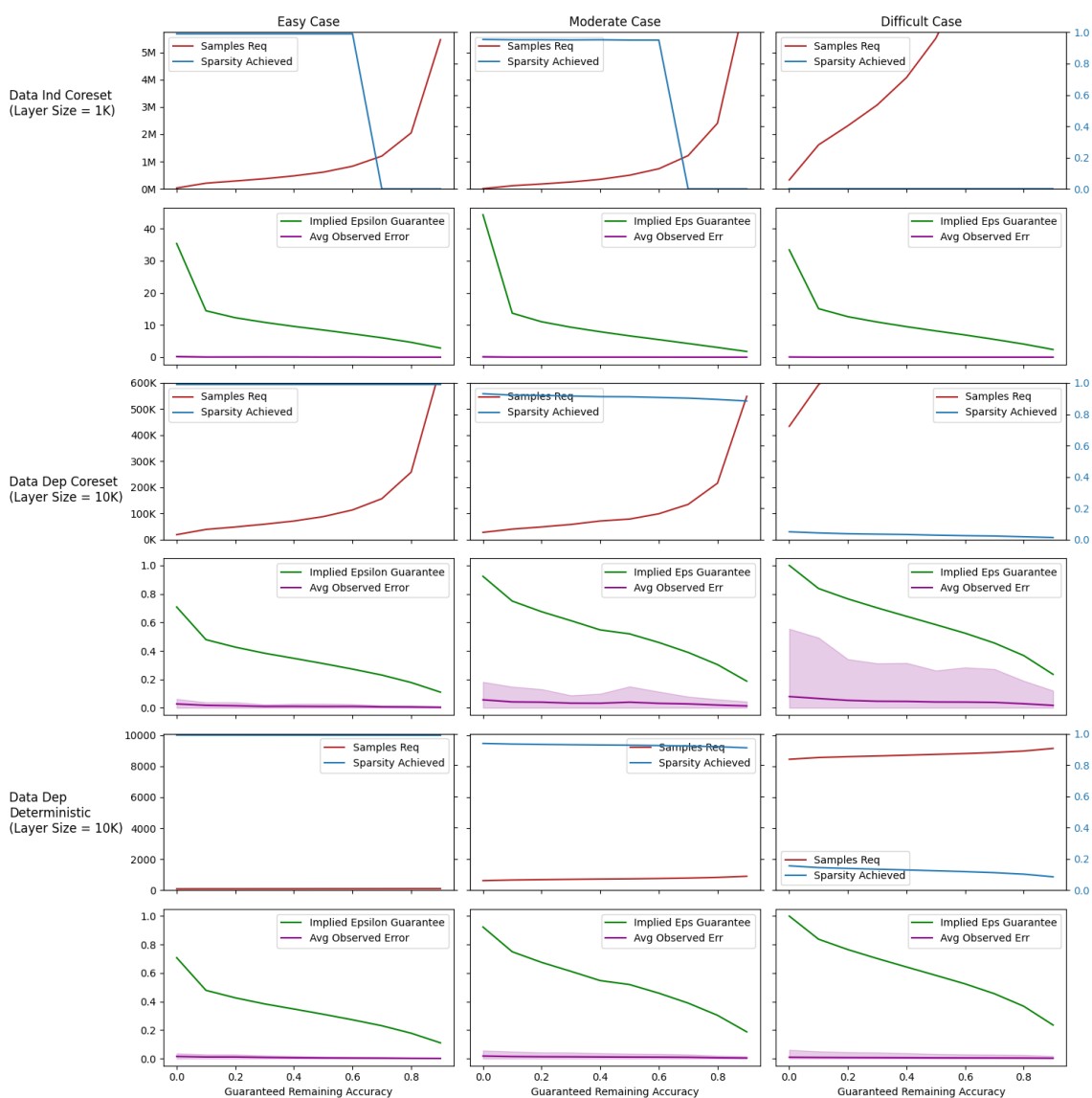

Figure 10: Full results of pruning the final layer of VGG-16. See Figure 6 and Section B.3. If the number of samples required is > 1M, we refrain from performing the computation and assume that the sparsity is 0. (Though in the easy and moderate cases, it would be > 0.)

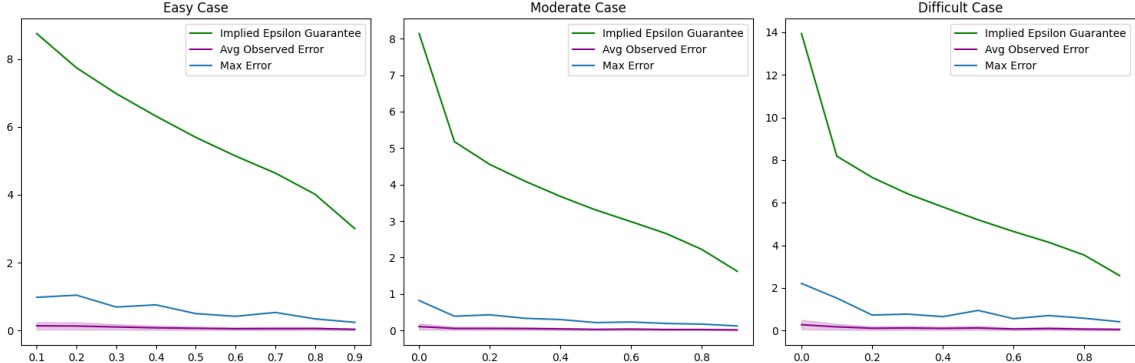

Figure 11: Approximation error of neuron pruning with a coreset method, when the inputs are not taken from the training distribution but are instead drawn from the L2 ball with radius $\beta$, which is the maximum L2 norm allowed by the assumptions of the approximation bound proof.

Proving label-consistency requires the *positive homogeneity* property of ReLU feed-forward neural networks (Du et al., 2018a), which says that multiplying a weight matrix by $\alpha$ results in scaling the output of the network by $\alpha$. The proof is very similar to the reasoning above. Let $f$ be the two-layer ReLU network with weights $(W^{(1)}, W^{(2)})$, and $\hat{f}$ be the network with weights $(\hat{W}^{(1)}, W^{(2)})$. For any input $x \in \mathcal{X}$, we have

$$\hat{f}(x) = W^{(2)}\phi(\hat{W}^{(1)}x) = W^{(2)}\phi(\alpha W^{(1)}x) = \alpha W^{(2)}\phi(W^{(1)}x) = \alpha f(x)$$

Finally, we can note that scaling the pre-softmax outputs by $\alpha$ does not change the predicted label:

$$\arg\max_i \hat{f}(x)_i = \arg\max_i \alpha f(x)_i = \arg\max_i f(x)_i$$

$\square$

A simple corollary of this idea is that we can arbitrarily increase or decrease the number of samples required to achieve *any* accuracy guarantee. From Lemma C.1, we know that scaling a weight matrix by a constant $\alpha$ increases the total sensitivity by $\alpha$. At the same time, it also *increases the additive epsilon bound required guarantee each label's consistency,* because the pre-softmax logits are multiplied by $\alpha$. This is equivalent to increasing or decreasing the temperature of the softmax distribution. Since sampling complexity grows roughly linearly with total sensitivity and shrinks quadratically with the size of the epsilon bound, the net result is that multiplying by $\alpha > 1$ *reduces* the required number of samples to achieve any particular accuracy guarantee.

**Corollary C.2** *In the context of Theorem A.4, multiplying the weight matrix $W^{(1)}$ by a positive constant $\alpha > 1$ decreases the number of samples required to achieve any particular accuracy guarantee.*

And indeed, we can see in Figure 12 that when we scale the layer weights of LeNet-300-100 up by $\alpha$ at various magnitudes, the additive epsilon bound required guarantee any accuracy grows proportionally, which quadratically shrinks the number of samples required to meet the guarantee.

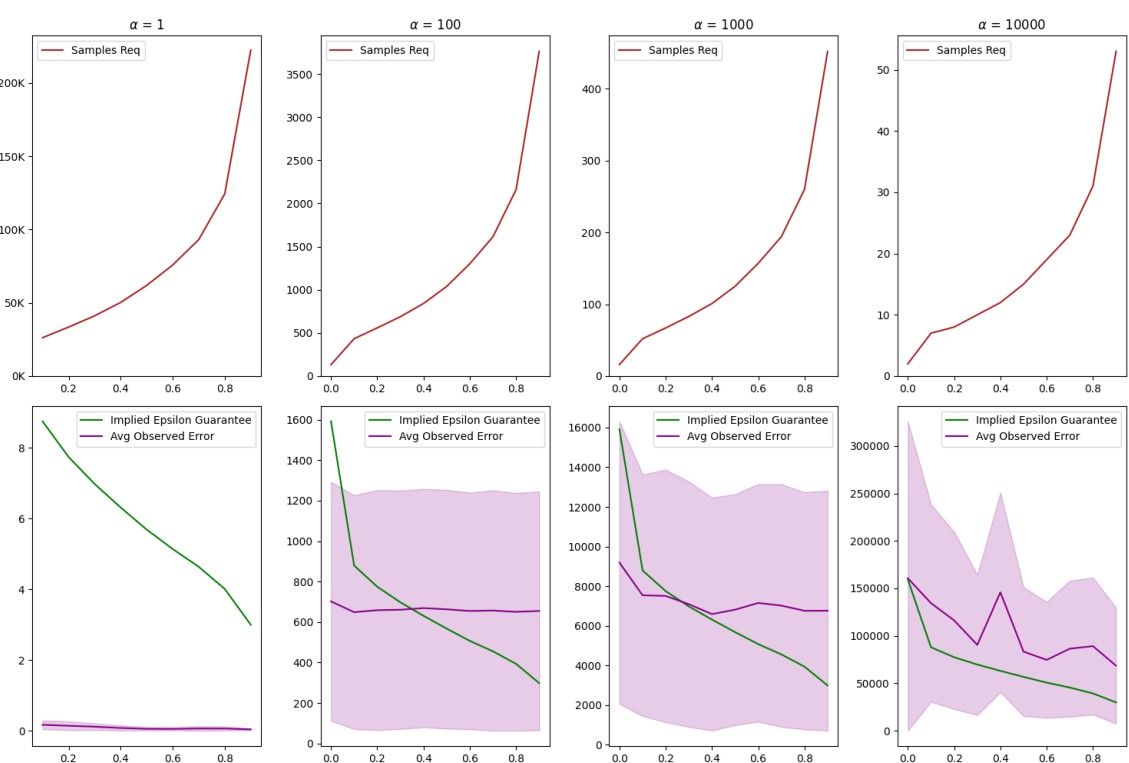

Figure 12: Results for pruning LeNet-300-100 with a coreset based neuron pruning method, after multiplying the pruned layer's weight matrix by a constant $\alpha$.

