# OpenReview forum: "When Are Neural Pruning Approximation Bounds Useful?"
_ICLR.cc/2021/Conference — Reject_

### Official Review · AnonReviewer2 · 2020-10-23
**Review of "When Are Neural Pruning Approximation Bounds Useful?"**

**Rating:** 5
**Confidence:** 4

**Review:**

The paper studies recent approximation bounds of coreset-based pruning strategies for neural networks.  It concludes that the bounds in certain cases can be very conservative (guarantee much lower accuracy than seen in practice), especially if the network can be fine-tuned after pruning,  and should not be used as a guide to estimate expected accuracy.

While the conclusion that the bounds can be conservative is not unreasonable -- that was not the point of the papers it tries to criticize, bounds do not have to be good estimates.  Furthermore, I find the paper to be poorly and confusingly written, with many questionable choices done in setting up error metrics, and the experiments (please see details below), so I recommend to reject it.

A bound on performance and an estimate of performance serve different purposes.  Even tight bounds that can not be improved may give poor estimates, but they are still useful in providing theoretical understanding of a problem, and possibly in safety-critical applications where one needs to guarantee worst case results.  So the criticism of the paper may be besides the point of the papers it analyzes.  They are among the first to show that coreset-based methods allow any kind of theoretical approximation guarantees, but they do not claim that the bounds are good predictors of performance.   The present paper would have more value if it attempted to improve the bounds, or perhaps develop accurate estimators of accuracy vs. pruning level, or maybe propose a pruning method which is more effective empirically.

Furthermore, for a given neural network that has been pruned it is fairly inexpensive to numerically evaluate the approximation accuracy w.r.t. to the full model on a validation set -- so I do not see much practical value of the exercise of trying to infer it from bounds.  Furthermore, for pruned networks that have later been fine tuned,  indeed the results of the papers do not apply, so the bounds unsurprisingly may have little to do with empirical results.

Detailed comments (non-exhaustive list of issues, but sufficient for my rating):

1) You go from additive and multiplicative bounds to error rates in lemma 3.1.  Just because epsilon > (f(x)i - f(x)i')/2  does not guarantee that it will make a mistake (it has the ability to make a mistake, but it does not have to).  Lemma 3.1. is on the one hand elementary, but on the other hand it introduces another layer of approximation between accuracy and pruning.
2)  "guaranteeing that 50% of the development set labels that are correct will remain correct"...   The goal of approximating a neural net is to make sure that the output of the approximate NN matches the original one -- how is it relevant whether the original was correct or not w.r.t. to the target labels?  Why do you condition on predictions being correct?
3) 'fine tuning can recover large amounts of accuracy while simultaneously maintaining of increasing approximation error" ... That doesn't make sense.  Do you mean 'improving' or 'reducing' approximation errors?
4)  Sections 2.1 (and same for 2.2).   A lot of important details are left to appendix -- this section gives very little useful information to understand the method.  Where is algorithm 1 -- is it in your appendix or is it in Baykal's paper?  Where are the theorems -- in your paper / appendix / Baykal's paper?  What is the 'importance', how is it computed? A more informative intro would be much appreciated.
5) The easy problem is a NN with 90 to 95% of weights set to 0.  The network is refitted, and 0-weights are re-enabled?  What does it mean for 0 weights to be re-enabled?  If the pruning strategy keeps the 0-edge -- does it mean the edge is active and the network is non-sparse?  This is very confusing (either poorly described, or this is an artificial corner case). It seems hard to imagine that a coreset pruning method that is given a sparse network will make it less sparse.
If on the other hand the accuracy bounds somehow ignore edges which are already 0 -- then it seems like an easy opportunity to derive better bounds?
6) For the above easy network (with mostly zero weights) -- the coreset algorithm gives more than 50% of elements with probability 0 of being sampled, so the conclusion is that if we ask for 50% sparsity -- it will run forever?  This whole discussion just seems very strange and confusing.
7) does 90% sparsity mean that there are 90% zeros or 90% non-zeros?  Maybe use clearer language.
8) The discussion of scaling up the bounds due to total homogeneity --  also seems very strange: the paper claims that positive homogeneity allows to 'arbitrarily increase or decrease sample complexity by scaling the appropriate weight matrices by a constant factor'.  Is this for the additive or multiplicative bound?  The paper says "investigating the full implications of this observation are out of scope of this paper".  I would expect some discussion here.  For the additive bound -- if you scale up the weigths in NN, the relative approximation error will be much smaller, so isn't it natural to require larger sample complexity?

---

> ### Author Response · Authors · 2020-11-11
> **Response to Review #2 (1/2)**
>
> Dear Reviewer #2,
>
> Thank you for taking the time to read and review our paper. While it is unfortunate that the motivation and content of our work is unclear to you, we hope the following comment will help alleviate some misunderstandings.

---

> > ### Author Response · Authors · 2020-11-11
> > **Response to Review #2 (2/2)**
> >
> > **“That’s not the point of these papers.”**
> >
> > We agree that pursuing a deeper understanding of pruning by trying to bound approximation error is a worthwhile endeavor. We also understand that coreset approximation bounds are simply a first step in this direction.
> >
> > However, that’s not how these papers are being marketed. Every one of the papers we reference (two were previously accepted at ICLR) prove a bound on the **pre-softmax** approximation error of a **single** layer neural network. Each of the papers go on to perform the following empirical analysis:
> >
> > 1. Prune a **multi-layer** neural network using sparsity-first pruning.
> > 2. **Ignore** the bounds on approximation error that were proven in the paper. (The bounds are never plotted!)
> > 3. **Fine-tune** the network (!) and compare the **development accuracy** of their method to other heuristic methods.
> >
> > The empirical results in these papers are misleading and are used to convince readers of two ideas:
> >
> > 1. Coreset methods achieve similar or better development accuracy than heuristic methods. This improvement is implicitly attributed to the direct approximation of the original network.
> > 2. Coreset methods are preferable because they have better guarantees. These guarantees can be used in proofs of generalization, layer-wise sparsity tuning, etc.
> >
> > Both of these “marketing” claims are directly contradicted by our work. When we correctly evaluate the approximation guarantees without the confounding factors of fine-tuning and multiple layers, we find that they are inappropriate for most of the proposed applications. When we add back in those confounding factors, we find that improvements in development accuracy are mostly explained by the properties of fine-tuning and not by the direct approximation of the pre-softmax logits.
> >
> > A paper should be judged based on its factual correctness and whether an audience at the conference will benefit from reading it. We believe our work meets both of these criteria: many practitioners are interested in applying these methods and may erroneously believe that they are suitable for guaranteeing performance after pruning. Others are interested in extending these coreset methods. Our work highlights several important challenges for the field, including the analysis of fine-tuning and the incorporation of softmax translation invariance into bounds.
> >
> > **“A bound on performance and an estimate of performance serve different purposes.”**
> >
> > We agree that a bound on worst-case performance serves a different purpose than an estimation of performance. But even the worst-case bounds are incredibly vacuous, equivalent to claiming “a mouse is no larger than a skyscraper.” While technically true, the bounds are not useful in practice, even for safety-critical applications. We show that requiring even mild guarantees on the accuracy of the network results in very small amounts of sparsity, even when we know that it’s possible to prune a large amount using heuristic methods while achieving a close approximation.
> >
> > **“It’s inexpensive to evaluate the validation accuracy. Why bother predicting it?”**
> >
> > This misses the point. Predicting validation accuracy is simply a *measuring stick* to help us evaluate how much approximation error is reasonable. Is an additive approximation error bound of 1 good or bad? An intuitive answer is that it depends on how much perturbation is required to flip X% of the development set labels, since ultimately label-consistency is what we care about.
> >
> > **“Confusingly written with questionable choices.”**
> >
> > Comment 3: It is correctly written that **fine-tuning increases approximation error**. We are aware that this is non-intuitive, which is why we dedicated the second half of the paper to demonstrating the phenomenon and explaining it. Coreset approximation bounds only provide guarantees on the pre-softmax output activations. Because of the translation invariance of softmax, this means that fine-tuning can increase the approximation error as measured by coreset approximation bounds.
> >
> > Comment 4: Yes, Algorithm 1 and more thorough explanations of Mussay and Baykal’s algorithms are located in the appendix. We were forced to move these introductions to the appendix due to the page limit.
> >
> > Comment 5: Yes, the “easy” case network is an artificial corner case to illuminate how pruning methods do even in the simplest of circumstances, when we know the network can be easily pruned while achieving 0 approximation error. We can think of it as a “debugging” input. A surprising result of this paper is that coreset methods require astronomical sample complexity even in this softball case.
> >
> > Comment 6: In sparsity-first pruning (which is different from epsilon-first pruning) we sample until we achieve X number of unique samples. If the number of elements with non-negligible probability is less than X, then this approach will not terminate because we can never sample X unique elements.
> >
> > Comment 7: 90% sparsity means 90% zeros.

---

> > > ### Comment · AnonReviewer2 · 2020-11-16
> > > **thanks for the clarifications!**
> > >
> > > Thank you for the well-written reply, I find the language and the explanations in the response much clearer than in the paper.  On the one hand I agree with the logic that overly loose bounds could be misleading in labeling one approach 'principled' and having a moral ground over other heuristic approaches,  while being comparable in performance.  On the other hand it's fairly common to analyze a highly simplified version of a proposed algorithm, as the version with full details is too complex to analyze.  Thanks for clarifying a couple of the confusing parts, where I may have missed the point you're making.
> > >
> > > I increase my rating of the paper, but I would find it much more convincing if either (i) you were able to give some improvements on the bounds, especially as in some cases they seem egregiously loose (already sparse network)  or (ii) show that coreset-based algorithms are not competitive in practice with alternative approaches or heuristics.

---

### Official Review · AnonReviewer4 · 2020-10-27
**The paper considers the trade-off between sparsity and fidelity by neural network pruning. It provides an empirically evaluation of the predictive power of two recently proposed methods based on coreset algorithms, and identify several circumstances in which the bounds are loose. It also examines the role of fine-tuning in prunability and observes that tight approximation bounds could be poor predictors of accuracy.**

**Rating:** 6
**Confidence:** 4

**Review:**

Overall, I vote for accepting. I like the idea of analyzing the practical performance of coreset-based pruning methods and discussing the relation between sparsity, approximation bounds, and accuracy.

Pros:
1. The paper provides an empirical evaluation that shows the theoretical bounds of coreset-based algorithms could be loose.  This is an important issue for the practicability of coreset-based algorithms.
2. The paper shows the tight approximation bound is not a necessary condition for accuracy by analyzing fine-tuning. The quantitative results that fine-tuning increases approximation errors are interesting.

Cons:
1. The paper regards the loose bounds as a shortage that weakens the predictive power of coreset-based pruning methods. But this can also be regarded as an advantage of coreset-based algorithms that can heavily increase sparsity while maintaining accuracy. I suggest the authors to give more discussions on both sides.
2. The experiments only consider coreset-based pruning methods since they have provably guarantees. However, due to the empirical results of fine-tuning, we can recover accuracy after post-pruning. Then it is interesting to also compare with other commonly-used pruning methods, e.g., compare the sparsity of different pruning methods that achieve the same accuracy with fine-tuning.

Some typos:
1. Figure 2 -> Table 2

---

### Official Review · AnonReviewer1 · 2020-10-30
**Good observation but a bit lack of novelty**

**Rating:** 5
**Confidence:** 3

**Review:**

This paper investigates when neural pruning approximation bounds useful from several perspectives. Showing that approximation bounds are loose and require a large sample size. It also explores the influence of fine-tuning after pruning by approximation bounds.

However, I a bit concerned about the novelty and impact of the paper. I cannot find border impacts and applications on the topic discussed in the paper since all results are based on observation. No further insights on tighter bounds or discussion of the reasons proposed. I would increase my score if those insights provided.

Also, in the last line of page 6, the author mentions Figure 4.2 which does not appear in the paper.

---

> ### Author Response · Authors · 2020-11-16
> **Response to Review #11**
>
> Dear Reviewer #1,
>
> Thank you for taking the time to review our paper. Below are a few comments to help clarify our submission.
>
> **Broader Impact and Applications**
>
> We’ve moved the “Related Work” section from the Appendix into the main body of the paper, which may alleviate your concerns about the broader impact and applications of our work. We discuss “over-parameterization” in that section, which is a common method of improving the trainability of neural networks by increasing the size of the network. Over-parameterization requires compression methods such as pruning to improve the efficiency of inference by removing redundancy in the discovered solution.
>
> As for pruning methods with *approximation bounds*, three potential applications of approximation bounds mentioned in *the literature* are given in Section 5.4. They are
>
> 1. Improving the sparsity/accuracy trade-off
> 2. Optimally tuning the sparsity of each layer
> 3. Aiding in proofs of generalization
>
> However, one of the main points of our work is that the first two applications are unlikely to benefit from tight approximation bounds; they will likely require a more thorough understanding of sparse fine-tuning dynamics. In spite of this, we believe that the ubiquity of neural network pruning makes accurately predicting the outcome of pruning inherently attractive, since the accuracy of our predictions serves as a measuring stick for our understanding of how pruning really works.
>
> **Providing Tighter Bounds**
>
> We agree that it would be nice if our work also provided tighter bounds. However, we believe this topic would be better communicated in a separate paper. Trying to squeeze the presentation, proof, and empirical evaluation of new methods into the 8 pages we have is unlikely to be readable. We address this topic more in our [Response to all Reviewers](https://openreview.net/forum?id=Xxli_LIvYI&noteId=nYCEGdz3lrS).

---

### Author Response · Authors · 2020-11-16
**Response to All Reviews**

Dear Reviewers and Area Chair,

We are extremely grateful for the time you have invested in reading and reviewing our work. Special thanks to Reviewer #2 for engaging with us during this rebuttal period.

**No New Methods**

We agree that our work would be more compelling if we also presented concrete solutions or new methods to address the challenges we highlight in our paper. However, our work suggests that improving the tightness of approximation bounds is not necessarily the correct goal for most of the pruning community. While some researchers interested in proofs of generalization may find tighter bounds useful, the majority of the pruning community would be better off focusing on the analysis of the dynamics of fine-tuning sparse models.

Also, given the depth and difficulty of these two problems, we believe that each topic deserves its own paper (or papers). Attempting to squeeze the presentation, proof, and empirical evaluation of any new theorems into the 8 pages we have is unlikely to be readable. While we do not present any new methods or SOTA results, we believe that our work, which is based on solid empirical evaluation and clear, honest communication has a place in the scientific discourse surrounding neural network pruning.

Regardless of the decision to accept or reject our paper, we again thank reviewers for graciously sharing their time with us.

---

### Decision · Program_Chairs · 2021-01-07
**Final Decision**

**Decision:**

Reject

**Comment:**

This work studies corset-based pruning strategies for neural networks, and highlights the looseness of approximation bounds, the difference between approximation error and probability, and the importance of considering post-pruning fine-tuning. I found the empirical findings and concerns raised around the utility of approximation bounds for pruning guarantees interesting and important, and appreciated the benchmark with varying levels of difficulty. However, the empirical analysis was limited to coreset-based methods and a simple LeNet architecture, and could benefit from considering additional non-coreset based approaches, architectures, and datasets. While I agree with the authors that new methods are not required for their work to be valuable, I believe that a more thorough empirical analysis is needed to support that their claims that current approximation bounds are not useful across wider experimental settings.